# Zinc-Dependent Histone Deacetylases in Lung Endothelial Pathobiology

**DOI:** 10.3390/biom14020140

**Published:** 2024-01-23

**Authors:** Rahul S. Patil, McKenzie E. Maloney, Rudolf Lucas, David J. R. Fulton, Vijay Patel, Zsolt Bagi, Anita Kovacs-Kasa, Laszlo Kovacs, Yunchao Su, Alexander D. Verin

**Affiliations:** 1Vascular Biology Center, Medical College of Georgia, Augusta University, Augusta, GA 30912, USA; 2Department of Pharmacology and Toxicology, Medical College of Georgia, Augusta University, Augusta, GA 30912, USA; 3Department of Cardiothoracic Surgery, Medical College of Georgia, Augusta University, Augusta, GA 30912, USA; 4Department of Physiology, Medical College of Georgia, Augusta University, Augusta, GA 30912, USA; 5Department of Medicine, Medical College of Georgia, Augusta University, Augusta, GA 30912, USA

**Keywords:** lung vascular endothelium, endothelial barrier integrity, zinc-dependent HDACs, deacetylation, HDAC inhibitors, acute lung injury, acute respiratory distress syndrome

## Abstract

A monolayer of endothelial cells (ECs) lines the lumen of blood vessels and, as such, provides a semi-selective barrier between the blood and the interstitial space. Compromise of the lung EC barrier due to inflammatory or toxic events may result in pulmonary edema, which is a cardinal feature of acute lung injury (ALI) and its more severe form, acute respiratory distress syndrome (ARDS). The EC functions are controlled, at least in part, via epigenetic mechanisms mediated by histone deacetylases (HDACs). Zinc-dependent HDACs represent the largest group of HDACs and are activated by Zn^2+^. Members of this HDAC group are involved in epigenetic regulation primarily by modifying the structure of chromatin upon removal of acetyl groups from histones. In addition, they can deacetylate many non-histone histone proteins, including those located in extranuclear compartments. Recently, the therapeutic potential of inhibiting zinc-dependent HDACs for EC barrier preservation has gained momentum. However, the role of specific HDAC subtypes in EC barrier regulation remains largely unknown. This review aims to provide an update on the role of zinc-dependent HDACs in endothelial dysfunction and its related diseases. We will broadly focus on biological contributions, signaling pathways and transcriptional roles of HDACs in endothelial pathobiology associated mainly with lung diseases, and we will discuss the potential of their inhibitors for lung injury prevention.

## 1. Introduction

Acute lung injury (ALI) and its more severe manifestation, acute respiratory distress syndrome (ARDS), are triggered by a variety of external stimuli, ranging from bacterial or viral pneumonia and inhalation injuries (direct injury) to sepsis (indirect injury). As a result, dysfunction of the lung alveolar–capillary barrier, hyper-inflammation, atelectasis, pulmonary edema, hypoxia, and ultimately death can occur [1,2,3,4]. Despite a 27–45% mortality rate [5], no pharmacological agents have been shown to improve outcomes in clinical trials in ARDS [1,6]. The current treatment guidelines rely on supportive management, including conservative use of fluids and airway maintenance, while controlling or resolving the inciting factor [1,2,7]. New therapies targeting epithelial–endothelial barrier preservation may reduce vascular leak and have profound clinical benefits.

Endothelial cells (ECs) form the lining of all blood vessels and create a semi-selective barrier between blood and interstitium. Endothelial dysfunction has been previously identified as a prominent risk factor in cardiovascular diseases [8] and ALI/ARDS [9,10]. Given the well-established understanding that SARS-CoV-2 or its spike protein causes endothelial dysfunction [11], COVID-19 is widely recognized as an endothelial disease primarily affecting the microvasculature [12]. Overall, preventing EC barrier integrity is a critical clinical requirement for treating various acute and chronic lung diseases [13,14,15].

Generally, the regulation of endothelial barrier integrity relies on an equilibrium between adhesive and contractile forces. Adhesive molecules present at the cell–cell and cell-matrix junctions provide EC barrier integrity and a scaffold for contractile machinery based on the actomyosin interaction; alterations at these junctions are primarily responsible for EC barrier disruption [16,17]. The tight interconnection between the endothelial cells is provided by the interaction of junctional proteins (occludins, zonula occludens-1 (ZO-1), catenins, and vascular endothelial cadherin (VE-cadherin), which are linked to the actin cytoskeleton of cells in its vicinity [18,19]. Consequently, any alterations in the dynamics of the actin cytoskeleton or/and the activation state of the EC contractile machinery can influence the stability of cell–cell junctions and may result in barrier dysfunction. Hence, the dysregulation of the endothelial barrier function is crucial (reviewed in [20,21]) in lung-related pathologies.

The tensegrity model proposed by Ingber and coworkers [22] highlights how the interconnected cytoskeletal components experience tension and compression, allowing cells to sense mechanical forces via specialized mechanoreceptors. Such mechanoreceptors can be found in the lungs and are known as rapidly adapting receptors [23]. Various mechanosensors, such as ion channels, integrins, focal adhesions, and cytoskeletal complexes, enable cells to respond to both external and internal mechanical perturbations. This mechanism is crucial in cell functions like spreading, migration and overall cellular responses. For instance, endothelial cells in the microvasculature receive mechanical stretch through integrins from the capillary wall, while focal adhesions link the actin cytoskeleton to the cell’s interface with the extracellular matrix. They are vital players in regulating the integrity of the EC barrier as they provide additional tethering forces. ECs require cell–cell contact and enrolment of VE-cadherin-based junctional protein complex to convert the stretch into signals that promote their proliferation [24] and to initiate mechanosensitive signaling cascades in the vascular endothelium [25,26].

Epigenetic mechanisms have recently been implicated as a significant regulator of endothelial barrier function [27,28,29]. Histone post-translational modifications (PTMs) allow ECs to respond to intra- and extracellular stimuli [21]. One of the most prevalent reversible post-translational protein modifications is acetylation, or more correctly, Nε acetylation, and is governed by two classes of regulatory enzymes: (i) lysine acetyltransferases (KATs), also known as histone acetyltransferases (HATs). KATs/HATs facilitate the transfer of an acetyl group to a Lysine (Lys) residue, and (ii) lysine deacetylases (KDACs) or histone deacetylases (HDACs) catalyze the removal of acetyl groups. In simplified terms, these enzymes add or remove acetyl groups from histones. Their mode of action is illustrated in Figure 1. HDACs can be classified into two subgroups: zinc-dependent histone deacetylases (Class I, IIa, IIb, and IV HDACs) and NAD^+^-dependent sirtuins (Class III HDACs). It has been shown that HDACs from both subgroups can regulate EC function at various levels and by diverse mechanisms [29,30,31]. Precise functional tuning of HDACs may be achieved by PTMs like phosphorylation, SUMOylation, etc. [32,33]. In addition to deacetylating histones, HDACs also deacetylate other non-histone proteins, which regulate cellular functions such as cytoskeletal polymerization and signal transduction [34,35]. A summary of the classification, structural features, and role of HDACs in EC function is provided in Table 1 [36,37,38,39,40,41,42,43,44,45,46,47,48,49,50,51,52,53,54,55,56,57,58,59,60,61,62,63,64,65,66,67]. The structural, functional, and inhibitory aspects of zinc-dependent HDACs have recently been reviewed by Porter and Christianson [68].

Compelling evidence suggests that dysregulation of protein acetylation can ultimately promote the emergence of several diseases, including ALI. Therefore, maintaining the balance between the activities of HATs and HDACs is critical for vascular homeostasis [29]. A recent review by Shvedunova and Akhtar explored various dimensions of acetylation and deacetylation, including their diverse targets, rapid turnover, and sensitivity to the concentrations of co-factors like acetyl-CoA, acyl-CoA, and NAD^+^, highlighting their close relationship with metabolism and homeostasis [69].

In addition to the direct de-acetylation of histones, HDACs play a role in epigenetic regulation as co-factors in many transcriptional processes [70,71]. Specific transcription factors mainly enlist them and are essential for maintaining vascular homeostasis and facilitating the development of blood vessels [72]. The dysregulation of HDAC expression and activity has been observed in numerous cardiovascular diseases like chronic obstructive pulmonary disease (COPD) [73], asthma [74], and cardiac hypertrophy [75]. Consequently, HDAC inhibitors have emerged as potential therapeutic targets to cope with these diseases [76].

A growing body of evidence indicates that inhibition of zinc-dependent HDACs leads to the preservation of EC barrier function and can be beneficial for treating cardiovascular and inflammatory diseases [27,77]. However, the roles of individual HDACs in EC barrier regulation in vitro and in vivo are largely unspecified. In this review focusing on vascular pathobiology (mainly in ALI, ARDS, angiogenesis, and CVD), we discuss and highlight the importance of zinc-dependent HDACs in the maintenance and regulation of endothelial barrier integrity, mechanisms, and core signaling pathways modulating endothelial cell functions. Further, we provide an overview of current HDAC research, and we evaluate possible therapeutic approaches and treatment strategies involving HDAC inhibitors to improve EC dysfunction.

## 2. Zinc-Dependent HDACs: Classification, Functions, Regulations and Modulations

To date, 18 mammalian HDACs have been discovered. Initially, they were divided into four classes based on their homology to yeast HDACs, but later on they were subdivided into two major groups: (i) “classical” zinc-dependent enzymes, which require zinc ion (Zn^2+^) as a co-factor for their catalytic function (classes I, II, IV) and (ii) sirtuins (class III), which are nicotinamide adenine dinucleotide (NAD^+^)-dependent enzymes [78,79]. This review will focus on the role of the “classical” HDACs in endothelial pathobiology. For this group, the presence of Zn^2+^ in the active site of HDAC provides an appropriate conformation for the catalytic domain, thus facilitating interaction and deacetylation of the substrate [80]. Zn^2+^ allows a nucleophilic attack of the H_2_O molecule on the acetyl group, thus promoting the removal of the acetyl moiety [81]. Therefore, the presence of Zn^2+^ is essential for the proper folding and stability of “classical” HDACs and their enzymatic activity [78,82]. The main molecular and structural characteristics of Zn-dependent HDACs are provided in Figure 2 and Appendix A.

In addition to their homology to yeast counterparts, “classical” zinc-dependent HDACs can be distinguished based on their structural features, subcellular locations and active mechanisms (Figure 2 and Appendix A). Class I HDACs (HDAC1, HDAC2, HDAC3, and HDAC8) have high homology to the yeast RPD3 gene [83,84,85] and are characterized by the presence of the conservative catalytic domain with N-and C-terminal extensions. The latter includes a Nuclear Localization Sequence (NLS) with regulatory phosphorylation sites (for HDAC1, HDAC2, and HDAC3) (Figure 2). These HDACs are ubiquitous enzymes that are primarily localized in the nucleus and have high catalytic activity against histones [85,86]. In addition, they can be involved in chromatin remodeling as an enzymatically active subunit of multi-protein complexes (functional interactomes), which include other HDACs and transcription factors [86,87]. Class II HDACs are homologous to the Hda1 gene [85,88] and can be present in both the nucleus and the cytoplasm. Compared to class I enzymes, they are more spatiotemporally regulated with more evident tissue and cell-restricted expression patterns [89,90] (Table 1). They are subdivided into class IIa and IIb subclasses (Figure 2). Class IIa HDACs (HDAC4, HDAC5, HDAC7, and HDAC9) are characterized by a unique N-terminal extension, which includes a binding site for the transcription factor MEF2, NLS, and three conservative phosphorylation sites. These sites regulate HDAC critically with a 14-3-3 protein chaperones, and 14-3-3 are responsible for cytoplasmic sequestration [91]. Additionally, class IIa HDACs have a C-terminal extension, which contains a Nuclear Export Signal (NES) (Figure 2). Therefore, class IIa HDACs can shuttle between the nucleus and cytoplasm. They primarily act as co-repressors of transcriptional events in the nucleus and have minimal deacetylase activity on acetylated histones [92,93,94]. However, cumulative evidence indicated they can deacetylase non-histone proteins [39,95] (Table 1). Class IIb HDACs (HDAC6 and HDAC10) possess two tandem deacetylase domains. However, in the case of HDAC10, the C-terminal (or Leu-rich) domain is truncated and lacks its enzymatic activity [96,97,98] (Figure 2). Unlike HDAC6, which is usually a cytoplasmic deacetylase, HDAC10 resides in both the nucleus and cytoplasm [97,98] (Table 1). HDAC11 is the only member of class IV HDACs and has a catalytic core homology with both class I and II enzymes. It is characterized by short N- and C-terminal extension and a unique ability to deacetylase fatty acids [99]. HDAC11 defatty-acylase activity is about 10,000 times more efficient than its deacetylase activity [100].

Initially, HDACs were discovered as enzymes that catalyze the removal of ε-N-acetyl groups from histones, modulating gene transcription (Figure 1). Later research revealed that HDACs can deacetylate non-histone proteins such as p53, signal transducers and activators of transcription (STAT3), E2F1, heat shock protein 90 (Hsp90), and NF-κB [101] (Table 1). The deacetylation process driven by HDACs executes chromatin compaction and transcriptional repression, while histone acetylation accelerates chromatin accessibility and triggers the activation of gene transcription [76] (Figure 1). The acetylation and deacetylation of proteins may impact the target proteins’ stability, activity, localization, and interactions [102]. The reversible protein acetylation may indirectly affect other PTMs through mechanisms like crosstalk between PTMs, protein-protein interactions, recruitment of modifying enzymes, etc. In short, zinc-dependent HDACs are evolutionally conserved enzymes that epigenetically regulate gene expression in numerous cellular pathways (reviewed in [81,103]). Reversible protein acetylation and other PTMs, like phosphorylation, methylation, ubiquitination, etc., are the core mechanisms regulating protein function [104]. A large body of evidence indicates that the vast majority of signaling cascades involved in cell cycle progression, proliferation, apoptosis, survival, differentiation, development, angiogenesis, and inflammatory actions are regulated by the zinc-dependent HDACs [82,105,106].

## 3. Zinc-Dependent HDACs in Endothelial Function

### 3.1. Class I HDACs: HDAC1, HDAC2, HDAC3, and HDAC8

Emerging evidence implied that zinc-dependent HDACs are actively involved in the function as well as in the dysfunction of the endothelial lining [107,108]. However, the role of specific HDAC subtypes remains incompletely defined and somewhat contradictory. For example, a recent study demonstrated that nuclear class I HDACs—HDAC1 and HDAC2, but not nuclear/cytoplasmic HDAC3—are involved in lung microvascular EC barrier dysfunction induced by lipopolysaccharide (LPS) via the suppression of Sox18 gene expression [109]. However, another study demonstrated that the pharmacological inhibition of HDAC1, HDAC2, and HDAC3 leads to EC barrier dysfunction in mice and increased lung vascular leak by suppression of EC Roundabout4 (Robo4) receptor expression [110]. In addition, published data demonstrated that specific inhibition of HDAC1, HDAC2, and HDAC3 induces F-actin stress fibers in Human Umbilical Vein ECs (HUVECs) and suggested that down-regulation of these HDACs is involved in *Bacillus anthracis* lethal toxin (LT)-induced HUVEC barrier dysfunction [111]. Further, one group demonstrated that the overexpression of HDAC2 in human aortic ECs mitigated the oxidized low-density lipoprotein (OxLDL)-mediated vascular dysfunction induced by endothelial Arginase 2 (Arg2). This occurred via the direct binding of HDAC2 with the Arg2 promotor [112], and protects against EC dysfunction and atherogenesis induced by oxidized lipids in mice [113]. A study from another group indicated that upregulation of HDAC2 promotes EC dysfunction in mice and humans via repressing the expression of Manganese superoxide dismutase (MnSOD), thus increasing oxidative stress [114]. In addition, pro-inflammatory cytokines like TNF-α and IL-1β induce inflammation and EC dysfunction, at least in part, via down-regulation of RNAse1 in an HDAC2-dependent manner in HUVECs [115]. Interestingly, the known EC barrier-enhancing lipid mediator, sphingosine-1-phosphate (S1P) [116], can bind to HDAC1 and HDAC2 and inhibit their enzymatic activity [117]. Thus, the involvement of class I HDACs in endothelial function may be agonist- and tissue-specific. Accordingly, a recent review highlighted the regulatory role of a class I HDAC, HDAC1, as an environmental sensor that drives functions of ECs like inflammatory and NO signaling, redox balance, and angiogenesis in an agonist-specific manner [40].

The nuclear/cytoplasmic enzymes HDAC3 and HDAC8 also regulate EC function in cell- and agonist-specific fashion. In human lung microvascular ECs (HL-MVECs), LPS modestly but significantly increases HDAC3 activity. Inhibition of HDAC 3 (in tandem with HDAC6) protects against LPS-induced EC barrier dysfunction in vitro and in vivo, apparently via acetylation and suppression of heat shock protein 90 (Hsp90) chaperone function and attenuation of RhoA activity [118]. Accordingly, the downregulation of HDAC3 protects blood-brain barrier (BBB) integrity in oxygen-glucose deprivation/reoxygenation states by promoting PPARγ activation [119] in brain ECs. It prevents BBB leakage in type II diabetic mice via activation of the nuclear factor erythroid 2 (NFE2)-related factor 2 (Nrf2) [120], which controls the cellular resistance to oxidants [121]. It has also been found that the mitigation of HDAC3 combats Type 2 diabetes mellitus-induced endothelial dysfunction via the Keap1-Nrf2-Nox4 signaling pathway [122].

However, HDAC3 plays a somewhat contrasting role in atherosclerosis development, which is tightly linked with endothelial dysfunction [123]. While HDAC3 is critical for maintaining endothelial layer integrity in the areas of disturbed flow in atherosclerosis [44], its up-regulation is involved in inflammatory responses in HUVECs and pharmacologic inhibition of HDAC3 mitigates the development of atherosclerotic lesions in mice [124]. Interestingly, distinct roles for class I and IIa HDACs in regulating gene expression and EC signaling in response to disturbed flow were noted [125], thus highlighting the functional differences between HDACs.

While the role of HDAC8 in EC function is mainly unknown, emerging evidence demonstrated the association of this enzyme with cytoskeletal proteins responsible for smooth muscle contractility, such as α-actin [126] and cortactin [127]. The latter protein actively participates in EC barrier regulation (reviewed by Bandela et al. [128]). However, the role of HDAC8-mediated cortactin deacetylation in EC barrier regulation remains to be elucidated.

### 3.2. Class IIa HDACs: HDAC4, HDAC5, HDAC7, and HDAC9

These HDACs have a more restricted expression patterns compared to class I HDACs and some unique structural and functional features (listed in Figure 2), which allow them to shuttle between the nucleus and the cytoplasm. They are recognized for their involvement in endothelial cell migration, angiogenesis, inflammation, and embryonic development [30,129,130]. Recently published data indicated that the pan-class IIa inhibitor, TMP269, improves EC barrier function in an LPS-induced murine ALI model in vivo and in HLMVECs in vitro [131]. Since this inhibitor selectively binds to the catalytic site of class IIa HDACs [132], this data suggested that the catalytic activity of this HDAC group may play a role in EC barrier regulation despite the minimal activity against acetylated histones [94]. The effect of TMP269 can be attributed to the activation of the Rho pathway, which may occur, at least in part, via de-repression of adapter protein ArgBP2 (Arg-binding protein 2) transcription [131]. However, the role of individual class IIa HDACs, as well as the role of nuclear/cytoplasmic shuttling, was not specified.

HDAC4, a “gold standard” in class IIa HDACs research, is linked to early stress response in pulmonary fibrosis. At the same time, HDAC2 is responsible for chronic progression, suggesting that these HDACs may regulate gene expression of pro-inflammatory cytokines and fibronectin in concert [133]. Additionally, by triggering autophagy, HDAC4 regulates vascular inflammation [134]. The activation of HDAC4 prevented endothelial dysfunction in diabetes [135]. A transcription factor, specificity protein-1, i.e., SP1, reduces intestinal barrier dysfunction, oxidative stress, and inflammatory response after sepsis by upregulating HDAC4 while inhibiting the high mobility group box one protein (HMGB1) expression [136]. Long noncoding RNA cancer susceptibility candidate 11 (lncRNA CASC11) alleviates ox-LDL-induced injury (inflammation and apoptosis) in coronary microvascular ECs, at least in part, by stabilizing HDAC4 [137]. Therefore, while the information on the involvement of HDAC4 in EC barrier regulation is limited, published data suggests that HDAC4 may have an anti-inflammatory and barrier-protective role in ECs.

Vascular endothelial growth factor and its receptor (VEGF and VEGFR, respectively) are primary drivers of angiogenesis in ECs [138,139]. There is a direct correlation between HDAC4 phosphorylation and VEGF signaling. The HDAC4 phosphorylation can improve the motility of ECs. Insights on phosphorylated HDAC4 in angiogenesis are provided by Liu et al. [140]. Nox4 oxidizes HDAC4 and enhances its phosphorylation; thus, this cascade facilitates proper tube formation by ECs [141]. HDAC4, HDAC5, and HDAC6 control VEGFR-2 expression [142], and its deacetylation in ECs directly impact EC function [143]. In a study conducted by Urbich et al., HDAC5 was recognized as a suppressor of angiogenic gene expression in HUVECs [144]. Consistently, HDAC5 impairs angiogenesis in scleroderma ECs [145]. Recent data indicated that HDAC5 may be involved in LPS-induced inflammatory responses in human pulmonary artery ECs (HPAECs) [146].

The literature suggests that HDAC7 is perhaps the most distinct of all class IIa HDACs [147]. For example, unlike other class IIa HDACs, it may primarily be localized in the cytosol in specific cell types [148]. Further, the pro- or anti-angiogenic function of HDAC7 is dependent upon its cellular localization [147]. The role of HDAC7 in endothelial permeability remains controversial. While a recent study implicated the involvement of HDAC7 in *E. coli*-induced ALI in mice [56], the classical work of Dr. Olson’s group demonstrated that HDAC7 maintains vascular endothelial integrity during embryogenesis via downregulation of the expression of matrix metalloproteinase 10 (MMP-10) [53]. The small interfering RNA (siRNA)-based knockdown of Filamin B can reduce VEGF-induced HDAC7 cytosolic accumulation, repressing MMP-10 and NUR 77 genes and inhibiting VEGF-mediated vascular permeability [149]. A peptide (7A) derived from HDAC7 was reported as a driver for vascular regeneration through phosphorylation of 14-3-3γ [150]. In addition, the maintenance of the vascular lumen is dependent on the phosphatase 2A/HDAC7/ArgBP2 pathway, which controls the endothelial cytoskeletal dynamics and cell-matrix adhesion in zebrafish and cultured HUVECs [151]. Data from the literature indicates that dephosphorylation of HDAC7 by myosin light chain (MLC) phosphatase (MLCP) regulates nuclear import of HDAC7 in thymocytes, suggesting the role of MLCP/HDAC7 crosstalk in cytoskeletal remodeling in non-muscle cells [152]. MLCP plays a vital role in EC cytoskeletal remodeling, leading to EC barrier strengthening [153,154]. However, the role of HDAC7 in regulating MLCP activity/EC barrier function is unknown. It was shown that ectopically expressed HDAC7 complexes with β-catenin and 14-3-3 retain β-catenin in the cytoplasm, thus inhibiting HUVEC proliferation [51]. On the contrary, a recent study demonstrated that upregulation of HDAC7 increased β-catenin acetylation at Lys 49 accompanied by decreased phosphorylation at Ser 45, thus facilitating its nuclear import and proliferation in lung cancer cell lines [155], suggesting a cell-specific role of HDAC7 in the regulation of cell growth.

While the role of HDAC9 in lung endothelial function is largely unknown, it was shown that it can worsen endothelial injury in a cerebral ischemia/reperfusion model via increasing inflammatory activity and endothelial barrier dysfunction [156]. HDAC9 was identified as a responsible factor since it triggered the IκBα/NF-κB and MAPKs signaling pathways, leading to the promotion of brain ischemic injury [157], and it was involved in brain EC injury associated with intracranial aneurism by repressing miR 34a expression [158]. Interestingly, HDAC9 may play a pro- or anti-inflammatory role, depending on the cell type investigated [159]. Therefore, the role of class IIa HDACs in endothelial function is complex and apparently tissue- and agonist-specific.

### 3.3. Class IIb HDACs: HDAC6 and HDAC10

HDAC6 is known for its effects on cytoskeletal dynamics and cell–cell junctions as well as for regulation of EC mechanosensing and permeability [160,161,162,163,164]. As demonstrated by studies conducted by Gao et al. [165] and Zhang et al. [166], HDAC6 controls cell motility by governing tubulin and actin networks. HDAC6 is directly deacetylated α-tubulin at Lys 40 [167], thus promoting MT disassembly in human pulmonary ECs [168]. However, it may affect MT dynamics independently from its deacetylase activity [169]. It is also demonstrated that HDAC6 plays a regulatory role in endothelial cell migration and angiogenesis via deacetylation of the actin-binding protein cortactin [60]. In addition, HDAC6 stimulates angiogenesis through polarization and migration of vascular ECs in a microtubule end-binding protein 1 (EB1)-dependent fashion [170]. The involvement of HDAC6 in EC barrier regulation is well documented due, at least in part, to the discovery of HDAC6-specific inhibitors (reviewed by Zhang et al., [171]). In many cases, downregulation or pharmacological inhibition of HDAC6 attenuates the increase in EC permeability induced by pro-inflammatory agonists or edemagenic microorganisms [118,168,172,173,174,175]. However, mechanisms are varied and may be agonist-specific. They may include microtubule destabilization accompanied by deacetylation of tubulin [168,173,174] and β-catenin, followed by disruption of adherens junctions [168] and activation of Rho-mediated contractile machinery [118,175]. In particular, *Staphylococcus aureus*-induced EC inflammation and barrier dysfunction is mediated by an increased level of reactive oxygen species (ROS), with subsequent HDAC6 activation followed by MT destabilization, which triggers the activation of the nucleotide exchange factor GEF-H1/Rho pathway [175]. In addition, recent data suggests that HDAC6 may activate Rho by either deacetylation (inhibition) of Rho GDP-dissociation inhibitor (RhoGDI) or upregulation of ArgBP2 expression [131,176]. ROS-mediated HDAC6 activation accompanied by HDAC6 phosphorylation at Ser 22 is involved in the disruption of EC barrier integrity induced by cigarette smoke [174]. Collectively, this data supported the multifaceted role of HDAC6 in EC barrier dysfunction. However, recently published data demonstrates that in primary HLMVECs derived from neonatal lungs and in a neonatal murine model of sepsis, downregulation of HDAC6 exacerbates LPS-induced pro-inflammatory responses via upregulation of canonical toll-like receptor (TLR) signaling, suggesting an anti-inflammatory role of HDAC6 in the development of lung pathology during systemic sepsis [177]. Therefore, the role of HDAC6 in various ALI models requires further investigation.

Along with HDAC6, HDAC10 plays an essential role in Hsp90-mediated proteasomal degradation and regulation of VEGF receptors [62]. A recent study demonstrated that HDAC10 promotes angiogenesis in HUVECs via the PTPN22/ERK axis [63]. In some cell types, up-regulation of HDAC10 is linked to pro-inflammatory responses [178,179]. However, the downregulation of HDAC10 exacerbates NLRP3-mediated inflammatory responses in intracerebral hemorrhage (ICH)-induced injury [180]. Further, a newly published study demonstrated that nebulized inhalation of HDAC10 attenuates oxidative stress, inflammation, and pulmonary fibrosis in a murine model of silicosis, suggesting an anti-inflammatory role of HDAC10 in lung injury [181]. The role of HDAC10 in lung EC barrier function remains to be elucidated.

### 3.4. Class IV HDAC: HDAC11

HDAC11 is the most recently discovered zinc-dependent HDAC [182] and the only member of class IV HDACs with some unique features (listed in Figure 2). The high expression of HDAC11 is associated with poor prognosis in non-small cell lung cancer [183]. HDAC11 is also critically involved in metabolic diseases, including obesity and diabetes (reviewed in [184]). In particular, acetaldehyde and the NF-κB pathways regulate the levels of HDAC11 in alcoholic liver disease. Further, the downregulation of HDAC11 decreased IL-10 levels, which increased TNF-α levels, thus promoting inflammation [185]. Accordingly, recent studies demonstrated that TNF-α increases the expression of HDAC11 in HUVECs, resulting in the activation of NLRP3/caspase-1/GSDMD and caspase-3/GSDME signaling cascades and culminating in pyroptosis, an inflammatory type of cell death involved in atherosclerosis [66,67]. Proteinase-activated Receptor-2 (PAR2)-induced HUVECs barrier compromise involves repression of VE-cadherin expression via upregulation of HDAC11 [65]. This data supported the involvement of HDAC11 in inflammatory responses and EC barrier compromise. Interestingly, the initial characterization of HDAC11 revealed that the enzyme co-immunoprecipitates with HDAC6, suggesting the formation of a functional complex [182]. However, the role (if any) of a potential HDAC11/HDAC6 interaction in EC function requires further investigation.

## 4. EC-Mediated Central Signaling Cascades and Their Regulation by Zinc-Dependent HDACs

Zinc-dependent HDACs contribute to vascular pathobiology through multiple mechanisms, such as impairing the production of nitric oxide (NO), increasing oxidative stress, triggering severe inflammatory cascades, differentiation, proliferation, apoptosis, altering integrity of ECs, modulating vascular tone and angiogenesis [29,30,186,187]. Broadly simplified links between HDACs and vascular regulatory mechanisms are presented in Figure 3. The engagement of class I HDACs in overall cellular proliferation, differentiation, and development is reviewed by Reichert et al. [188]. Class I and II HDACs were studied in ECs with regard to inflammation, oxidation, and proliferation-related signaling cascades and mRNA, i.e., gene expression studies, and interpreted as prominent therapeutic targets in EC dysfunction [125].

In particular, HDACs can epigenetically modulate the activity of regulators (cyclins, cyclin-dependent kinases, p53, etc.) involved in cell cycle progression and proliferation [189,190]. Under pathological circumstances (e.g., stress, injury, hypoxia, etc.), ECs tend to migrate and proliferate, thus disturbing EC barrier integrity [191,192]. HDAC1 to HDAC7 are active participants in these pathomechanisms [107]. Accordingly, HDAC inhibitors, such as SAHA (suberoylanilide hydroxamic acid, vorinostat) [193], trichostatin A (TSA), apicidin [194], and tubacin [195], were reported to overcome barrier dysfunction and restrict EC proliferation. Interestingly, in cancer cell lines, HDAC3 and HDAC8 downregulate the expression of SIRT7, thus facilitating proliferation [196,197] and suggesting a possible crosstalk between zinc-dependent and NAD^+^-dependent (class III) HDACs in the regulation of cell proliferation.

In apoptosis and cell survival pathways, HDACs affect the expression of pro-apoptotic and anti-apoptotic genes, thus maintaining the equilibrium between cell death and survival signals [198]. Apoptosis has been linked to several cardiovascular diseases, and a positive correlation between an increased EC apoptosis and the development and progression of cardiovascular diseases has been established [199,200,201,202]. In a HUVEC model of disturbed flow (atherosclerosis), β-catenin positively regulates endothelial nitric oxide synthase (eNOS) activity and anti-apoptotic gene expression [203]. In turn, β-catenin activity and cellular location are controlled by HDAC7 in HUVECs, suggesting the involvement of HDACs in β-catenin-mediated EC proliferation and survival [51]. HDAC1 and HDAC2 modulate the expression of endothelial VCAM-1 and atherogenesis by mechanisms that downregulate the methylation of the GATA6 promoter by blocking STAT3 acetylation [204]. HDACs are considered an epigenetic factor in the modulation of vascular endothelium-related atherosclerosis and flow [28]. MicroRNA-200b-3p can induce EC apoptosis in atherosclerosis by encountering HDAC4 [205]. The involvement of HDACs in atherosclerosis, emphasizing the regulation of endothelial cell function, is reviewed by Chen et al. [107].

Cellular differentiation, development processes, and angiogenesis (VEGF and its receptors) are also epigenetically controlled by HDACs [198]. Hrgovic et al. explained that the anti-angiogenic action of HDAC inhibitors like TSA, sodium butyrate (But), and valproic acid (VPA) by VE-cadherin-dependent mechanism suppresses VEGFR-2 expression [142]. HDAC1, HDAC4, HDAC5, and HDAC6-mediated suppression is a crucial target for anti-angiogenic drug development.

Inflammatory events promote injury, vascular dysfunction, and a breach in EC barrier integrity. Most of the time, the inflammation and its regulation are HDAC-dependent [107,206,207]. In particular, HDACs modulate the expression of genes encoding inflammatory cytokines, chemokines, and adhesion molecules, thereby affecting the inflammatory cascades, including the NF-κB pathway [206,208,209,210]. In 2019, Bedenbender et al. elucidated how inflammatory cytokines hindered the equilibrium of endothelial RNase1-eRNA homeostasis, which buffers against different vascular pathologies [115]. They reported that the inflammatory stimulation of ECs facilitates HDAC2 activation, which subsequently leads to H4 and H3K27 deacetylation (within the promoter region of *RNASE1*). The vascular architecture disturbs the vascular inflammation (TNF-α) driven by HDAC1, HDAC2, HDAC3, HDAC5, HDAC6, HDAC7, HDAC8, HDAC9, and HDAC11 [30,107,211]. The functional roles of HDACs in CVD-related inflammations are reviewed by Kulthinee et al. [212].

ROS, oxidative stress, and uncoupling of endothelial nitric oxide synthase (eNOS) are the critical components for inducing endothelial dysfunction [213]. HDAC2, HDAC3, and HDAC6 are associated with oxidative stress/ROS production [211]. HDAC4/HDAC5-HMGB1 mediated signaling, facilitated by the activity of NADPH oxidase, plays a vital role in cerebral ischemia/reperfusion injury or ischemic stroke [214]. HDACs like HDAC1, HDAC2, HDAC3, HDAC5, and HDAC6 and their inhibitors were shown to modulate NO production [215].

## 5. Therapeutic Targeting of Zinc-Dependent HDACs in Lung Injury

The broad relationship between zinc-dependent HDACs and their inhibitors is shown in Figure 4. A deeper understanding of the structural features, selectivity and capability of various HDAC inhibitors was reviewed by Roche and Bertrand [216]. These authors have summarized the advancements made by chemists in designing more specific and effective compounds and distinguished their behavior within target molecules. Briefly, HDAC inhibitors are subdivided into four classes based on their chemical structure: hydroxamate, cyclic peptide, benzamide, and aliphatic acids (Figure 4). TSA, belinostat, panobinostat, and vorinostat (SAHA) belong to the hydroxamate class. Among these, pan-HDAC inhibitors, such as TSA and vorinostat, are canonical zinc-dependent HDAC inhibitors that inhibit HDAC1 to HDAC9 with about equal potency [217]. Depsipeptide belongs to the cyclic peptide class. Inhibitors classified as benzamides, MS-275 (etinostat) and MGCD0103 (mocetinostat) are usually considered to target HDAC1, HDAC2, HDAC3, and HDAC8 (class I HDACs). 4-Phenylbutyric acid (phenylbutyrate), sodium butyrate (sodium salt of butyric acid), and valproic acid (valproate) are characterized as aliphatic acids (Figure 4). In general, zinc-dependent histone deacetylase inhibitors share similar pharmacophores that consist of three essential components: a cap group, a linker, and a zinc-binding domain.

The potential role of HDAC inhibitors as therapeutic agents in the treatment of a wide variety of diseases has been established (reviewed by Li et al. [218] in 2019 and Li et al. [219] in 2022). HDACs have recently been suggested as promising potential targets for treating cardiac diseases [220]. HDAC inhibitors are also reviewed in treating sepsis and inflammatory pulmonary diseases [221,222]. Table 2 summarizes broad knowledge of the role of HDAC inhibitors in lung pathobiology [46,66,142,223,224,225,226,227,228,229,230,231,232,233,234,235,236,237,238,239]. Emerging evidence demonstrated that ALI/ARDS is regulated (not entirely but at least partially) by HDACs [240,241,242]. However, the information on the involvement of specific HDAC-mediated signaling pathways in ALI/ARDS is limited and based primarily on the effect of semi-selective HDAC inhibitors on ALI manifestations in murine models.

It was demonstrated that treatment with a broad-spectrum inhibitor of zinc-dependent HDACs, valproate [243], administrated 30 min or 3 h after *Escherichia coli* (*E. coli*) infection attenuates pro-inflammatory responses in mouse lungs. However, administration of the same inhibitor at 4 h, 6 h, or 9 h after *E. coli* did not affect lung injury, suggesting a limited therapeutic window for this treatment [224,225]. Accordingly, the therapeutic index of valproate is reported to be narrow in both mice and humans due to side effects and toxicity [244].

Recently, Kasotakis and co-workers [56] reported that TSA can effectively mitigate lung inflammation and improve the survival rate of mice infected with *E. coli*. Surprisingly, the *E. coli*-induced injury in murine lungs is accompanied by decreased transcription of several HDAC genes, i.e., HDAC2, HDAC7, and HDAC8, but increased the expression of HDAC7 at the protein level. Further, TSA selectively decreased HDAC7 mRNA after an *E. coli* insult and attenuated *E. coli*-induced HDAC7 at the protein level. These discrepancies are discussed in more detail in the form of a letter to the editor [245]. Notably, the mechanisms by which TSA attenuates lung injury in *E. coli*-induced murine pneumonia are unclear. TSA was suggested to attenuate lung injury induced by mechanical ventilation/bleomycin insult in mice via inhibition of HDAC4/Akt signaling. However, TSA selectivity towards HDAC4 has not been described [246]. TSA and SAHA attenuate ventilator-induced lung injury in rats [247]. In addition, literature data indicated that another broad-spectrum HDAC inhibitor, sodium butyrate (SB), attenuates LPS-induced lung injury in mice [248]. Both TSA and SB are effective against intrapulmonary inflammation and promote survival in murine sepsis [249].

The nuclear HDAC1 and HDAC2 participation in the endothelial barrier compromise in vitro and lung injury in an LPS murine model was recently evaluated [109]. Selective inhibition of HDAC1 and HDAC2, but not HDAC3, attenuates LPS-induced hyper-permeability in HLMVECs, likely through the preservation of Sox 18 expression, which LPS downregulates. Further, the HDAC1 inhibitor, tacedinaline, preserves Sox 18 expression and attenuates LPS-induced ALI in mice. Therefore, this study suggested that HDAC1 and HDAC2 act as transcription repressors, which drive the downregulation of Sox18 transcription induced by LPS, thus compromising the integrity of the pulmonary EC barrier. Therefore, selective HDAC1/HDAC2 inhibition may benefit ALI/ARDS treatment.

A recent study suggested that the chemical constituent of *Azadirachta indica*, Nimbolide, inhibits LPS-induced ALI in mice via suppression of oxidative stress and inflammatory responses, at least, in part, via inhibition of nuclear translocation of NF-κB and HDAC3 in alveolar epithelial cells [250]. Another study reported that selective inhibition of HDAC3 by the RGFP966 compound suppresses the expression of pro-inflammatory cytokines (such as IL-6, IL-1β, and IL-12β) in macrophages via inhibiting activity of NF-κB and attenuated lung injury manifestations in murine lung slices treated with LPS/IFNγ [251]. Interestingly, HDAC3 in macrophages may regulate the expression of pro-inflammatory genes in vitro and in mice in a dichotomous manner, with HDAC3 activity suppressing LPS-induced signaling. At the same time, the recruitment of HDAC3 by LPS activates the expression of pro-inflammatory genes independently from HDAC3 catalytic activity [252]. Therefore, specific precautions should be taken to generate HDAC3-specific inhibitors for treating lung inflammation and injury by targeting HDAC3 enzymatic activity [253]. While combined inhibition of HDAC3 and HDAC6 (by RGFP-966 and tubastatin, respectively) attenuates LPS-induced ALI in mice [118], a recent study demonstrated that MS-275 compound (entinostat), inhibits HDAC1, HDAC2, and HDAC3 activities [254], suppresses Robo4 expression by inhibiting HDAC3 in ECs and enhances EC permeability and lung vascular leakage in mice [110].

Selective HDAC6 inhibition by tubastatin attenuates LPS-induced deacetylation of α-tubulin and β-catenin in the lung, which was accompanied by reduced pulmonary edema [168], suggesting the therapeutic value of HDAC6 inhibition in ALI treatment.

The role of zinc-dependent HDACs in managing chronic lung diseases is more complex. Pathological cell type-specific imbalance of class I and II HDAC activities was observed in idiopathic pulmonary fibrosis (IPF). Positive outcomes have been predicted in managing IPF through HDAC1 and HDAC2 inhibition by Kim et al. [255]. This group recently reported HDAC1 as a signature marker in systemic sclerosis-associated interstitial lung disease [256]. However, the pan-HDAC inhibitor, SAHA, is not practical in the treatment of cystic fibrosis ex vivo [257]. Further, HDAC2 is essential for suppressing pro-inflammatory gene expression in chronic lung diseases like COPD and severe asthma [258,259].

Zinc-dependent HDACs (mainly HDAC3, HDAC4, and HDAC8) are suggested as potential therapeutic targets in managing hypertension [260]. It was also reported that specific HDAC6 inhibition with tubastatin A attenuated lung injury in a rat model of COPD [261], thus offering a promising therapeutic strategy for COPD treatment. Further, there is a significant demand for the development of isoform-specific inhibitors for the treatment of lung injury as well as other diseases [262]. As a therapeutic remedy option, it is also essential to revisit and check the efficacy and selectivity of existing inhibitors. In addition, drug repurposing can be another therapeutic option to cure lung injury. For example, it was recently demonstrated that escitalopram, a selective serotonin reuptake inhibitor (SSRI), can be utilized as a treatment option for ALI. It inhibits the SIK2/HDAC4/NF-κB signaling pathway and overcomes the lung injury [263]. Combinational therapy, which includes HDAC inhibitors and DNA methylation modifiers, has been proven to be effective in murine models of LPS-induced inflammatory lung injury [230]. Further, developing dual-specificity inhibitors is another promising approach for treating lung injury and co-morbid pathologies. Thus, a recently designed dual inhibitor of HDACs and the PI3K/AKT pathway, CUDC-907, effectively treated bleomycin-TGFβ1-induced lung fibrosis as well as TGFβ1-induced tumor fibrosis [264].

## 6. Conclusions

As detection methods for post-translational protein modifications improve, the repertoire of cellular acylated molecules continues to expand. This necessitates discovering and validating responsible enzymes involved in these processes, as well as the functions of different targeting sites. Our current understanding of these modifications is primarily based on studies of a limited number of lysine/histone acetyltransferases and lysine/histone deacetylases. Additionally, understanding the interplay between epigenetics, post-translational modifications, and metabolism in different cell types presents a significant challenge. However, with the advancement of genetic and biochemical tools, future research will focus on unraveling the molecular specificity of zinc-dependent HDACs. This review is primarily focused on the role of zinc-dependent HDACs in the regulation of EC-mediated lung function, emphasizing that the role of these enzymes can be isoform- and agonist-specific. However, additional studies are required to further elucidate the role of specific zinc-dependent HDAC subtypes and individual enzymes in the complex process of lung function regulation.

## Figures and Tables

**Figure 1 biomolecules-14-00140-f001:**
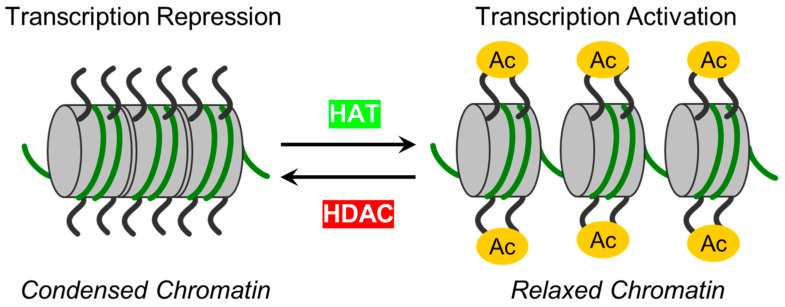
The schematic illustrates enzymes responsible for reversible protein acetylation in the regulation of gene transcription. Two classes of regulatory enzymes, lysine acetyltransferases (KATs)/histone acetyltransferases (HATs) and lysine deacetylases (KDACs)/histone deacetylases (HDACs), govern the reversible post-translational Nε acetylation of Lys residues in proteins. While KATs/HATs facilitate the addition of acetyl group to Lys residues and promote chromatin unfolding (relaxed chromatin), thus facilitating the activation of transcription, KDACs/HDACs catalyze acetyl group removal from histone and non-histone targets and are responsible for chromatin condensation (repression of transcription).

**Figure 2 biomolecules-14-00140-f002:**
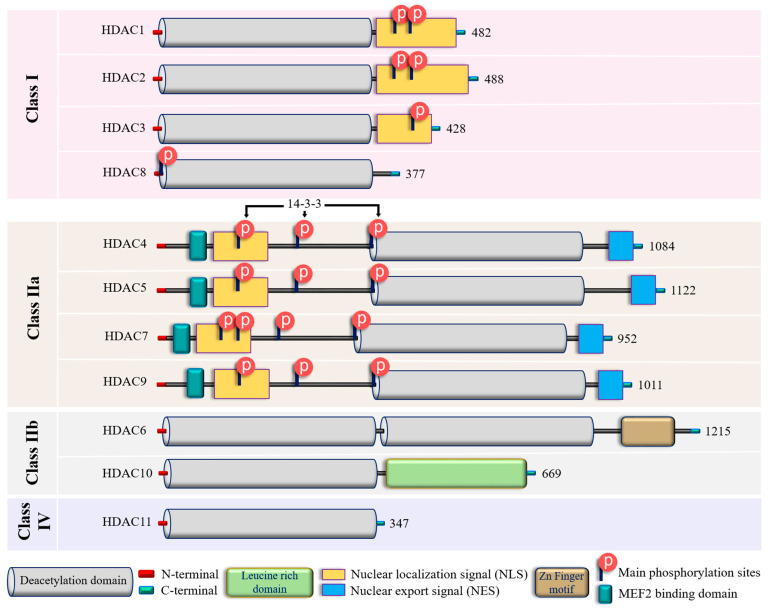
Structural features of zinc-dependent HDACs. The schematic depicts the main structural peculiarities of zinc-dependent HDACs. See the text for further explanation. Additional information on HDACs secondary and tertiary structures is provided in Appendix A.

**Figure 3 biomolecules-14-00140-f003:**
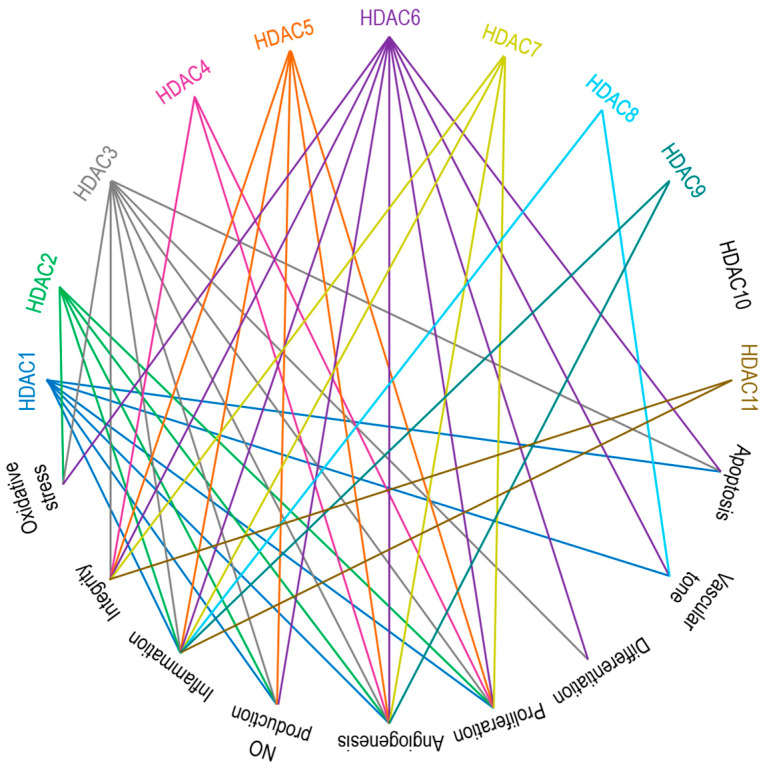
Main endothelial signaling pathways mediated by zinc-dependent HDACs. HDAC1, HDAC3, and HDAC6 are involved in apoptosis. HDAC1, HDAC6, and HDAC8 are responsible for vascular tone. HDAC3 and HDAC6 control cell differentiation. HDACs 1 to 7 participate in cell proliferation. Except for HDAC8, HDAC10, and HDAC11, all other zinc-dependent HDACs drive angiogenesis. HDAC1, HDAC2, HDAC3, HDAC5, and HDAC6 regulate NO production. Aside from HDAC4 and HDAC10, the rest of the zinc-dependent HDACs are involved in inflammatory responses. HDACs 3 to 7 and HDAC11 are the key players accounting for the regulation of cell integrity. HDAC2, HDAC3, and HDAC6 are contributors to oxidative stress. See the text for further explanations and references.

**Figure 4 biomolecules-14-00140-f004:**
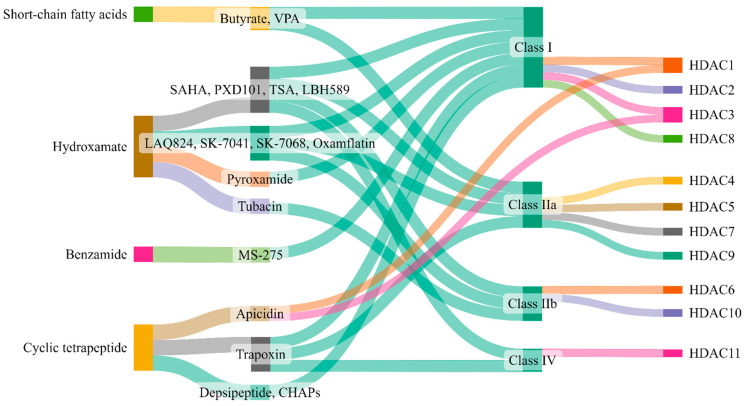
Zinc-dependent HDACs and their inhibitors. This figure (made at SankeyMATIC.com) illustrates four distinct classes of HDAC inhibitors: aliphatic acids, hydroxamate, benzamide, and cyclic peptide. Among these, trichostatin A (TSA), belinostat, panobinostat, and vorinostat are hydroxamate inhibitors, with TSA being the most extensively studied. Depsipeptide falls within the cyclic peptide class. Notably, the inhibitors MS-275 (etinostat) and MGCD0103 (mocetinostat) target HDAC1, HDAC2, HDAC3, and HDAC8, classified as class I HDACs, and are categorized as benzamides. Aliphatic acids encompass 4-phenylbutyric acid (phenyl butyrate), sodium butyrate (sodium salt of butyric acid), and valproic acid (valproate).

**Table 1 biomolecules-14-00140-t001:** Classification, structural features, and functions of zinc-dependent HDACs in vasculature.

Class	Isoform	Subcellular Distribution	Preferential Expression	A.A. Length	Non-Histone Substrates	Activities/Functions in Vasculature
Class I	HDAC1	Nucleus	Ubiquitous	482	p53, SHP, MyoD, STAT3, E2F1, AMPK, NF-kB, RB1, CtIP, ATF4, SRF [36,37,38,39]	Facilitates the impact of external and environmental stimuli on ECs [40]
	HDAC2	Nucleus	Ubiquitous	488	YY1, BCL6, GCCR, STAT3 [37]	Protect against DNA damage response and the onset of cellular senescence [41], critical for vascular homeostasis and endothelial health [42]
	HDAC3	Nucleus/ Cytoplasm	Ubiquitous	428	YY1, SHP, p65, GATA1, MEF2D, STAT3, ATF4, SUMO-LXR [37,38,39,43]	Preserves endothelial integrity [44]; Controls lung alveolar macrophage development and homeostasis [45]
	HDAC8	Nucleus/ Cytoplasm	Ubiquitous	377	Actin, SMC3 [37]; KMT2D, NCOA3, TUBA1A [39]	Culprit in hypertension [46]

Class IIa	HDAC4	Nucleus/ Cytoplasm	Heart, SM, Brain	1084	HP1, GATA1 [37]; SRF, ATF4, SUMO-LXR [38]; human transcription factor HIF-1α [39]	Regulates cellular senescence, apoptosis and autophagy, acts as inflammatory mediator [47,48] and a regulator of vascular endothelial growth factor D [49]
	HDAC5	Nucleus/ Cytoplasm	Heart, SM, Brain	1122	HP1, SMAD7 [37]; p53 [39]	Controls activity of KLF2, KLF2 activation in ECs; induces eNOS expression resulting in vasodilation [50]
	HDAC7	Nucleus/ Cytoplasm	Heart, Placenta, Pancreas, SM	952	PLAG1, PLAG2 [37]; HIF-1α [38]	Suppresses EC proliferation [51], controls EC proliferation and migration [52], maintains vascular integrity in embryogenesis [53], promotes promyelocytic leukemia protein sumoylation [54], Promotes angiogenesis [55]; involves in *E. coli*-induced ALI [56]
	HDAC9	Nucleus/ Cytoplasm	SM, Brain	1011	NA	Inflammatory mediator [57]

Class IIb	HDAC6	Cytoplasm	Heart, Liver, Kidney, Pancreas	1215	HSP90, SHP, SMAD, α-tubulin [37], G3BP1 [58]; Survivin, AKT, β-catenin, Peroxiredoxin, MMP-9 [38]; p53, ERK1, human cortactin [39]	Crucial in EC function [59], Regulates EC migration and angiogenesis [60], Important in atherosclerosis [61] and HSP90-mediated VEGFR regulation [62]
	HDAC10	Cytoplasm	Liver, Spleen, Kidney	669	AKT, β-catenin, MMP-9 [38]; N-acetylputrescine, N^8^-acetylspermidine [39]	Accelerates angiogenesis in EC via PTPN22/ERK axis [63], Pulmonary hypertension [64], Regulates HSP90-mediated VEGFR [62]

Class IV	HDAC11	Nucleus	Brain, Heart, SM, Kidney, & Testis	347	MyoD [38]; SHMT2 [39]	Compromises the vascular endothelial barrier function [65], Key player in atherosclerosis [66], Triggers caspase-mediated pathways (NLRP3/caspase-1/GSDMD; caspase-3/GSDME) causing pyroptosis [67]

HDAC—Histone deacetylase; A.A.—amino acids; SHP—Src homology 2 domain-containing phosphatase; MyoD—Myoblast Determination Protein; STAT3—Signal Transducer and Activator of Transcription 3; E2F1—E2F Transcription Factor 1; AMPK—AMP-activated protein kinase; NF-kB—Nuclear Factor-kappa B; RB1—Retinoblastoma protein 1; CtIP—CtBP-interacting protein; ATF4—Activating Transcription Factor 4; SRF—Serum response factor; YY1—Yin Yang 1—BCL6—B-cell lymphoma 6; GCCR—Glucocorticoid receptor; GATA1—GATA-binding factor 1; MEF2D—myocyte enhancer factor 2D; SUMO-LXR—Sumoylation Liver X receptor; SMC3—Structural Maintenance of Chromosomes 3; KMT2D—human histone-lysine N-methyltransferase 2D; NCOA3—human nuclear receptor coactivator 3; TUBA1A—human tubulin alpha-1A; HP1—Heterochromatin Protein 1; SM—Skeletal muscle; SMAD7—Small mothers against decapentaplegic homolog 7; PLAG1—Pleomorphic adenoma gene 1; PLAG2—Pleomorphic adenoma gene 2; NA—Not available; HSP90—Heat shock protein 90; SMAD—Small Mother Against Decapentaplegic; G3BP1—Ras GTPase-activating protein-binding protein 1; MMP-9—Matrix metalloproteinase-9; SHMT2—Human serine hydroxymethyl transferase 2; KLF2—Krüppel-Like Factor 2; eNOS—endothelial Nitric Oxide Synthase; ALI—Acute Lung Injury; VEGFR—Vascular Endothelial Growth Factor Receptor; PTPN22—Protein Tyrosine Phosphatase Non-Receptor Type 22; ERK—Extracellular Signal-Regulated Kinase; NLRP3—NLR family pyrin domain containing 3; GSDMD—Gasdermin D; GSDME—Gasdermin E; LECs—lung endothelial cells.

**Table 2 biomolecules-14-00140-t002:** Zinc-dependent HDACs inhibitors in lung pathobiology.

Class	Inhibitor	Mode of Action	Reference/s
Class I	Valproic acid	Attenuate parameters of lung injury like oxidative stress, apoptosis, and inflammation, enhance HO-1 activity (ALI)	[223]
		Reduces levels of IL-6 and tumor necrosis factor (ALI)	[224]
		Reduces neutrophil influx into lungs and local tissue destruction via decreasing myeloperoxidase activity. Ameliorate pulmonary as well as systemic inflammatory response (ARDS)	[225]
		Antagonizes the inflammatory damage of vascular tissues	[226]
		Inhibits VEGFR-2 protein expression in angiogenesis	[142]
		Increases histone acetylation in thrombopoiesis	[227]
	Sodium butyrate	Inhibits VEGFR-2 protein expression in angiogenesis	[142]
	Trichostatin A	Alleviates HDAC4-mediated vascular inflammatory responses in hypertension	[228]
		Prevents I/R injury-induced activation of gene programs that include cell death and vascular permeability	[229]
		Inhibits VEGFR-2 protein expression in angiogenesis	[142]
	Trichostatin A + 5-Aza 2-deoxycytidine	Inhibits the eNOS-Cav1-MLC2 signaling pathway, enhances acetylation of histone markers and improves EC permeability (ALI)	[230]
		Reduce inflammation and promote an anti-inflammatory M2 macrophage by inhibiting MAPK-HuR-TNF and activating STAT3-Bcl2 pathways (ALI)	[231]
	miR-23b (HDAC2)	Reduces levels of IL-1β, IL-6, and TNF-α and inhibit HDAC2 (ALI)	[232]
	PCI34051 (HDAC8)	Reduces blood pressure via attenuating a component of the RAS or modulating nitric oxide signaling pathways (Hypertension)	[46]

Class IIa	Valproic acid	Same as Class I	-
	Sodium butyrate	Same as Class I	-
	Trichostatin A	Same as Class I	-
	TMP 195	Limits proinflammatory responses in Atherosclerosis	[233]
	MC1568	Abolishes NO-induced formation of macromolecular complexes and regulates downstream gene expression	[234]
	Tasquinimod (HDAC4)	Allosterically binds to HDAC4 and prevents HDAC4/nuclear receptor corepressor (N-CoR); HDAC3 complex formation, which inhibits HDAC4-regulated histone deacetylation and transcription, thus reduces inflammation in Angiogenesis	[235,236]

Class IIb	Sodium butyrate	Same as Class I	-
	Trichostatin A	Same as Class I	-
Class IIb (HDAC6)	CAY10603	Prevent ɑ-tubulin deacetylation, protects against inflammation, blocks IĸB phosphorylation, and reduces caspase-1 activation, particularly in epithelial cells (ALI)	[237]
	Tubastatin A	Inhibit angiotensin II-induced hypertension via protecting cystathionine γ-lyase protein degradation	[238]

Class IV	Trichostatin A	Same as Class I	-
	Hydroxytyrosol acetate	Inhibits pyroptosis by possible targeting of HDAC11 in TNF-α-induced HUVECs (Atherosclerosis)	[66]
	Quisinostat	Aggf1 regulates the pathophysiology of vascular endothelium; therefore, HDAC11 inhibitors restore the expression of Aggf1 in vascular injury	[239]

HO-1—Heme oxygenase-1; IL-6—Interleukin-6; ALI—Acute Lung Injury; ARDS—Acute Respiratory Distress Syndrome; VEGFR-2—Vascular Endothelial Growth Factor Receptor-2; HDAC—Histone Deacetylase; I/R—Ischemia/Reperfusion; eNOS—endothelial Nitric Oxide Synthase; Cav1—caveolin1; MLC2—myosin light chain 2; EC—Endothelial cell; MAPK—mitogen-activated protein kinase; HuR—Human Antigen R; TNF—Tumor Necrosis Factor; STAT3—Signal Transducer and Activator of Transcription 3; Bcl2—B-cell lymphoma 2; IL-1β—Interleukin-1 beta; TNF-α—Tumor Necrosis Factor Alpha; RAS—renin-angiotensin system; NO—nitric oxide; N-CoR—Nuclear Receptor Co-Repressor; IĸB—Inhibitor of κB; HUVECs—human umbilical vascular endothelial cells; Aggf1—Angiogenic Factor with G Patch and FHA Domains 1.

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
