# Peer review of "Zinc-Dependent Histone Deacetylases in Lung Endothelial Pathobiology"

_biomolecules, 2024, doi:10.3390/biom14020140_

Round 1
Reviewer 1 Report
Comments and Suggestions for Authors
The manuscript "Zinc-dependent Histone Deacetylases in Lung Endothelial Pathobiology" is an interesting review elucidating the possible role of HDAC inhibitors for endothelial barrier dysfunction and lung injury. The manuscript collects the large literature body available with over 250 citations. It describes in detail the HDAC role, classification, function, and dependency on Zinc for their proper functionality. The manuscript offers an interesting platform for the further evaluation and experimentation of HDAC inhibitors for ALI and CLI. One comment:
1. I encourage the authors to modify figure 4. The figure is hard to read and confusing. The authors should employ a different type of graph to link inhibitors and their HDAC targets (e.g. sankey charts).
Author Response
Reviewer #1:
I encourage the authors to modify Figure 4. The figure is hard to read and confusing. The authors should employ a different type of graph to link inhibitors and their HDAC targets (e.g. Sankey charts).
Ans. The figure has been modified according to the suggestion.
Reviewer 2 Report
Comments and Suggestions for Authors
Review Report for Manuscript biomolecules-2827905
The manuscript is very detailed in looking at Zinc-dependent Histone Deacetylases and their role in the pathobiology of lung diseases.
I suggest the authors to provide a graphical presentation within section 3. (Zinc-Dependent HDACs in Endothelial Function), that will present all the roles and impacts that different classes of HDACs have in endothelial functions.
Author Response
Reviewer #2:
I suggest the authors to provide a graphical presentation within section 3. (Zinc-Dependent HDACs in Endothelial Function), that will present all the roles and impacts that different classes of HDACs have in endothelial functions.
Ans. We agree with the reviewer that the overall graphical presentation will be useful. However, we would like to emphasize that the main message of the review is that HDACs roles in the endothelial function are incompletely understood and may be agonist tissue- and isoform-specific. Further, some data of the literature are controversial. Therefore, it is difficult to combine all existing data of literature in one figure. With respect, we prefer not to do it to avoid confusion and misleading interpretation. In addition, the results are summarized to some extent in Table 1.